# Selective Neck Dissection and Survival in Pathologically Node-Positive Oral Squamous Cell Carcinoma

**DOI:** 10.3390/cancers11020269

**Published:** 2019-02-25

**Authors:** Shunichi Shimura, Kazuhiro Ogi, Akihiro Miyazaki, Shota Shimizu, Takeshi Kaneko, Tomoko Sonoda, Junichi Kobayashi, Tomohiro Igarashi, Akira Miyakawa, Tadashi Hasegawa, Hiroyoshi Hiratsuka

**Affiliations:** 1Department of Oral Surgery, Sapporo Medical University School of Medicine, South 1 West 16, Chuo-ku, Sapporo 060-8543, Japan; s.shimura@me.com (S.S.); ogi@sapmed.ac.jp (K.O.); shimizushota1222@gmail.com (S.S.); t-kaneko@sapmed.ac.jp (T.K.); jkoba@sapmed.ac.jp (J.K.); t-igarashi@sapmed.ac.jp (T.I.); miyakawa@sap-oral.or.jp (A.M.); hiratuka@sapmed.ac.jp (H.H.); 2Department of Public Health, Sapporo Medical University School of Medicine, South 1 West 16, Chuo-ku, Sapporo 060-8543, Japan; tsonoda@sapmed.ac.jp; 3Department of Surgical Pathology, Sapporo Medical University School of Medicine, South 1 West 16, Chuo-ku, Sapporo 060-8543, Japan; hasetada@sapmed.ac.jp

**Keywords:** oral squamous cell carcinoma, selective neck dissection, survival, metastatic nodal parameters, pathologically positive nodes

## Abstract

The most important prognostic factor in oral squamous cell carcinoma (OSCC) is neck metastasis, which is treated by neck dissection. Although selective neck dissection (SND) is a useful tool for clinically node-negative OSCC, its efficacy for neck node-positive OSCC has not been established. Sixty-eight OSCC patients with pN1–3 disease who were treated with curative surgery using SND and/or modified-radical/radical neck dissection (MRND/RND) were retrospectively reviewed. The neck control rate was 94% for pN1–3 patients who underwent SND. The five-year overall survival (OS) and disease-specific survival (DSS) in pN1-3 OSCC patients were 62% and 71%, respectively. The multivariate analysis of clinical and pathological variables identified the number of positive nodes as an independent predictor of SND outcome (OS, hazard ratio (HR) = 4.98, 95% confidence interval (CI): 1.48–16.72, *p* < 0.01; DSS, HR = 6.44, 95% CI: 1.76–23.50, *p* < 0.01). The results of this retrospective study showed that only SND for neck node-positive OSCC was appropriate for those with up to 2 lymph nodes that had a largest diameter ≤3 cm without extranodal extension (ENE) of the neck and adjuvant radiotherapy. However, the availability of postoperative therapeutic options for high-risk OSCC, including ENE and/or multiple positive lymph nodes, needs to be further investigated.

## 1. Introduction

Oral squamous cell carcinoma (OSCC) constitutes a broad range of tumors with diverse etiologies that are classified by region. The estimated age-standardized rate is relatively large at 2.7 per 100,000. This rate is 3.7 among men and 1.8 among women, with an estimated 202,000 patients newly diagnosed with oral cancer in 2012 [1]. It is widely accepted that OSCC metastasizes, which usually occurs through the lymphatic system to the cervical lymph nodes, with the most important prognostic factor being the presence of neck metastasis [2,3,4,5]. Therefore, ablation surgery of the primary OSCC with appropriate neck dissection is considered to be the standard treatment modality. Indeed, some authors have shown that the failure pattern among OSCC patients is approximately 30%, which is typically due to regional metastases [6,7]. Thus, pathologically proven lymph node metastases (pN+) are recognized as an adverse prognostic factor in OSCC.

Most surgeons support comprehensive neck dissection, such as modified radical neck dissection (MRND) or radical neck dissection (RND), for the basic management of regional disease in patients with clinically positive node (cN+) involvement [8,9]. On the other hand, selective neck dissection (SND) removes lymph node groups at designated anatomic levels. Therefore, SND results in improved quality of life and a lower likelihood of orofacial or shoulder dysfunction compared to comprehensive neck dissection [10,11]. In OSCC, SND (of levels I–III) is generally accepted as a regional treatment option because OSCC mainly drains into neck levels I, II and III [6,9,12]. Therefore, for the management of regional lymph nodes in OSCC, SND (I–III) has become the standard elective procedure for clinically node-negative (cN0) patients or those with microscopic disease [13,14]. With respect to the management of cN+ OSCC, some authors agree that SND (I–III) may be considered together with adequate removal of the gross tumor with or without adjuvant radiotherapy [15,16,17]. However, whether therapeutic SND (I–III) is appropriate for cN+ disease in the neck remains unclear because the objective measures for patient selection have not been clearly defined [17].

The aim of this retrospective study was to evaluate the efficacy and outcomes of SND (I–III) for pN+ OSCC. In addition, we examined the prognostic significance of node-related factors in pN+ OSCC in order to evaluate SND (I–III) for pN+ OSCC.

## 2. Results

### 2.1. Outcomes and 5-Year Survival in Patients Who Underwent Neck Dissection

Recurrences at primary, regional and distant sites were demonstrated in 37 (28%), 11 (8%) and 3 (2%) of 131 patients who underwent neck dissection, respectively. The five-year overall survival (OS) was 67% for 131 patients: 80% for cN0, 68% for cN1 and 54% for cN2. Furthermore, the five-year OS was 77% for patients with pN0 and 55% (*p* = 0.01) for patients with pN+ during the follow-up interval, which ranged from 28 to 154 months with a median follow-up time of 77 months. Similarly, the five-year disease-specific survival (DSS) was 88% for patients with pN0 and 61% for patients with pN+ (*p* < 0.01).

Of the 131 patients who received neck dissection as the definitive surgery, the pathologic examination of the neck dissection specimens revealed metastatic lymph nodes in 68 patients (52%). Ipsilateral, bilateral and contralateral metastases were found in 54 patients, 13 patients and 1 patient, respectively (Table 1). With a minimum follow-up duration of 42 months for survivors in 68 patients with pN+, 26 patients (38%) died of OSCC and 7 (10%) died from other causes.

### 2.2. Regional Control and Prognostic Factors in 35 Node-Positive OSCC Patients Underwent SND (I–III)

Among the 68 patients with pN+ disease, 35 patients (51%) underwent SND (I–III), 28 patients underwent MRND and 5 patients underwent RND. The distribution of ipsilateral and contralateral metastasis according to the extent of neck dissection is shown in Table 2. The frozen section diagnoses of level III lymph nodes did not detect micrometastasis in 5 necks.

A total of 10 patients developed regional recurrence: 4 of these had undergone selective neck dissection (4/35, 11%), 5 had undergone MRND (5/28, 18%) and 1 had undergone RND (1/5, 20%). Regional recurrences were well controlled as this occurred only in two patients who underwent SND (Table 3). The five-year OS and DSS of the 35 pN+ OSCC patients who underwent SND were 62% and 71%, respectively (Table 4).

The number of lymph nodes resected ranged from 6 to 42 (mean, 18) and the number of positive lymph nodes was 1–5 (mean, 2). When the cutoff point for metastatic lymph nodes was set at 2, there were 30 patients with ≤2 and 5 patients with >2. The number of positive lymph nodes significantly affected OS and DSS (*p* < 0.01, Figure 1). Microscopic extranodal extension (ENE) was present in 4 patients, with no significant effect on OS and DSS in the univariate analysis (*p* = 0.52 and *p* = 0.98, respectively). The total number of excised lymph nodes was neither associated with OS nor DSS. The lymph node density (LND) was calculated as the ratio of positive lymph nodes to the total number of lymph nodes removed. The mean LND was 0.11 (range 0.01–0.33). When the cutoff value for high and low LND was set at 0.11, based upon the mean LND, a high LND was also not correlated with low OS and DSS (*p* = 0.06 and *p* = 0.10, respectively) in the univariate analysis.

In 35 pN+ patients who underwent SND (I–III), the number of positive nodes and surgical margin status were associated with lower survival rates for both OS and DSS in the univariate analysis. The final stepwise selection in the Cox proportional hazards regression model revealed that the number of positive nodes was an independent predictor of outcome for patients who received SND (I–III) (hazard ratio (HR) = 4.98, 95% confidence interval (CI): 1.48–16.72, *p* < 0.01 for OS; HR = 6.44, 95% CI: 1.76–23.50, *p* < 0.01 for DSS) (Table 5).

## 3. Discussion

The current retrospective comparative study that examined SND (I–III) and MRND/RND confirmed that only SND (I–III) with (4 patients) or without (31 patients) adjuvant radiotherapy for OSCC with pN+ that were limited to levels I and II was appropriate for patients with up to 2 tumor-positive lymph nodes with a largest tumor diameter of ≤3 cm rather than the immunohistochemical invasive tumor patterns or density of tumor–infiltrating CD8^+^ T cells. Some authors have suggested that SND (I–III) combined with adjuvant radiotherapy is a management strategy with high efficacy and minor morbidity for selecting pN+ OSCC patients [16,17,18]. However, most of these conclusions were not based on a comparison of SND (I–III) and MRND/RND. In addition, the selected patients were not appropriately defined. In a meta-analysis comparing SND with MRND/RND in OSCC patients with cN+ disease, Liang et al. [19] also suggested that cN+ OSCC patients treated with SND (I, I–III or I–IV) that was followed by adjuvant therapy had comparable clinical outcomes to MRND/RND because no significant difference was found for regional recurrence, OS or DSS between any of the dissection treatment types. However, the extent and selection of the SND levels differed between studies rather than being limited to the dissected fields of I–II. Moreover, the ipsilateral neck recurrence rates ranged from 8–15% for the SND group and from 3–16% for the MRND/RND group. Similar ipsilateral neck recurrence rates were found in the current study (11% for the SND group; 14% for the MRND/RND group), with regional control rates of 94% for the SND (I–III) group and 86% for the MRND/RND group. These results indicate that SND (I–III) could achieve good regional control and had a favorable prognosis for pN+ OSCC in the present cohort. For head and neck squamous cell carcinoma, including OSCC, Schmitz et al. [20] suggested that postoperative irradiation is not justified for pN1 neck disease without microscopic ENE.

Nodal fixation and gross ENE are considered as contraindications for SND as highlighted by Rodrigo et al. [21]. Thus, ENE is a predictor for worse prognosis of OSCC, while ENE does not serve as an independent prognostic factor [17], as shown in the present study. In particular, microscopic ENE is generally not easily detected by either physical examination or various imaging techniques.

In a series of 400 OSCC patients with primary surgery combined with neck dissection, Rogers et al. [22] reported that OS and DSS among 175 patients with pN+ disease had rates of 36% and 52%, respectively, compared to 67% and 87% among 314 patients with pN0 OSCC (*p* < 0.01 and *p* < 0.01, respectively). The outcomes for the 63 pN+ OSCC patients in the current study, with five-year OS of 57% and DSS of 63%, are comparable with the aforementioned cohorts. The major reason for the good survival of pN+ OSCC patients in the current study was the favorable outcomes for those with delayed neck development. It is well known that the adverse outcomes associated with delayed neck dissection can partly be explained by the fact that patients developing delayed cervical lymph node metastasis have a more advanced nodal stage. In our previous publications, we also postulated that the mode of invasion was associated with poor prognosis in patients with OSCC and with delayed cervical lymph node metastasis in patients with cN0 OSCC [23,24,25]. Therefore, after these results were obtained, we adopted a follow-up strategy of watchful waiting, especially for patients having grades 4C and 4D modes of invasion with node-negative OSCC, who were followed up carefully at short intervals in order to detect and treat delayed cervical lymph node metastasis as early as possible. By doing this, we could achieve favorable survival for patients with pN+ OSCC. Moreover, lymph nodes are vital immunological organs as they are essentially information marketplaces where antigen-presenting cells, the body’s scouts, come to display information that they have gathered regarding antigens that they have encountered in the field and patrolling lymphocytes find that specific antigens have entered the body [26]. Based on these anatomic physiological aspects, we chose a treatment policy for the avoidance of routine elective neck dissection and SND (I–III) was adopted mainly as a therapeutic procedure rather than an elective modality of management for cN0.

The current study revealed that the number of positive nodes was an independent predictor for both OS and DSS in the SND (I–III) group.

Givi et al. [27] also reported that the number of positive nodes (≤2 vs. >2) was a predictor of poorer two-year disease-specific survival in head and neck cancer.

Although confirming the results of previous studies [28,29,30] and supporting the widely held belief that the number of tumor-positive lymph nodes is associated with the outcome, this observation does not help in treatment planning before definitive surgery. Six of the 17 patients with 3 or more positive lymph nodes developed ipsilateral regional recurrence and died of OSCC. Five of these patients had not received postoperative radiotherapy. Although the long-term follow-up results from the RTOG 9501/Intergroup Phase III Trial suggested that head and neck cancer patients with 2 or more positive cervical lymph nodes did not benefit from adjuvant chemoradiotherapy [31], the availability of postoperative therapeutic options for high-risk OSCC, including multiple positive lymph nodes, needs to be further investigated.

Recent advances in cancer immunology have validated the concept that immune cells have potential key roles for effective cancer treatment [32]. Several chemotherapeutic agents and radiotherapy can induce immunogenic modulation [33]. Based on this information, combined adjuvant therapies using chemoradiation and/or immunotherapies, for which there is already evidence of efficacy for head and neck cancer, including OSCC, such as peptide vaccines [34,35] and the anti-programmed death 1 monoclonal antibody [36], may improve the prognosis of OSCC patients with multiple cervical positive lymph nodes.

## 4. Materials and Methods

### 4.1. Patients

From January 2004 through December 2014, inpatients who underwent definitive surgery with neck dissection, including SND (I–III) or MRND/RND, at the Department of Oral Surgery, University Hospital, Sapporo Medical University for either cN+ or cN0 OSCC were screened for this retrospective study. Although the neck dissection performed was dependent upon the site, size and extent of the primary tumor and nodal disease, which was diagnosed by clinical signs, including physical examination, and imaging techniques, such as computed tomography (CT) scanning, magnetic resonance imaging, ultrasonography (US), positron-emission tomography (PET)-CT and serum tumor markers, the extent of neck dissection was decided based on the following criteria. According to the results of our previous study [25], OSCC patients having up to 2 positive nodes limited to levels I and II, in addition to cN0 OSCC with delayed nodal development, underwent SND (I–III). MRND was indicated for OSCC with cN+ at levels III, IV and V. RND was indicated for OSCC with clinical ENE. Postoperative adjuvant chemotherapy and/or radiotherapy were performed for OSCC for those with at least 3 positive nodes and/or ENE, except for patients who refused additional treatment.

Of 221 OSCC patients who had surgical treatment, 131 patients received a neck dissection (93 ipsilateral, 37 bilateral and 1 contralateral), with 108 patients receiving this procedure at the same time as the initial surgery while the remaining 23 patients had delayed neck dissection without recurrence of the primary tumor. The types of ipsilateral neck dissection administered were: SND (I–III), 82 necks; MRND, 43 necks; and RND, 5 necks. The following procedures were used to treat the contralateral neck: SND (I–III) in 32 necks; and MRND in 6 necks. When a level III positive node was confirmed under the frozen section diagnosis during SND (I–III), the neck dissection was converted to MRND. Finally, the types of neck dissection performed were: SND (I–III), 114 necks; MRND, 49 necks; and RND, 5 necks.

None of the patients received previous treatment, except for neoadjuvant chemotherapy or chemoradiotherapy. Patients were excluded from the current study if they had received any form of neck dissection with resection of a recurrent primary OSCC. The tumor extent and the histopathological grading were classified according to the American Joint Committee on Cancer/International Union against Cancer (AJCC/UICC)-TNM staging system [37,38]. Follow-up evaluations were carried out with physical examination and imaging studies as mentioned above. The follow-up rate was 94.7% and data from patients lost to follow-up were censored.

This retrospective study was conducted according to the principles stated in the 1964 Declaration of Helsinki and its subsequent revisions, which was approved by the Institutional Review Board of our university on 12 September 2017 (no. 292-1116). Informed consent or an acceptable substitute was obtained from all patients before study inclusion.

### 4.2. Immunohistopathologic Analysis

Immunohistochemistry was employed to detect multi-cytokeratins, identify the tumor cells and determine CD8^+^ T-cell density. Briefly, 4 μm serial sections from paraffin-embedded samples deparaffinized in xylene were soaked in 10 mM citrate buffer (pH 8.0) and placed in an autoclave at 121 °C for 10 min for antigen retrieval. Endogenous peroxidase was blocked by incubation with 0.3% hydrogen peroxide in methanol for 30 min. After this, the sections were incubated with a primary monoclonal antibody targeting pancytokeratin (1:200, clone AE1/AE3, abcam, Cambridge, UK) and CD8 (Clone C8/144B, Code 413201, Nichirei Bioscience, Inc. Tokyo, Japan) at 4 °C overnight. Secondary antibodies were applied as indicated for the EnVision system (EnVision^+^; DAKO, Glostrup, Denmark). Staining was visualized with diaminobenzidine tetrachloride. The sections were counterstained with hematoxylin, dehydrated, cleared and mounted. Following this, serial sections from the paraffin-embedded samples were stained with hematoxylin and eosin for the assessment of the mode of invasion and the worst pattern of invasion (WPOI). The presence and absence of lymphovascular invasion and perineural invasion were tabulated according to the information from the routine histopathological reports.

Tumor budding was defined as the presence of isolated single tumor cells or a small group of fewer than five cells ahead of the deep invasive tumor parts. For the assessment of tumor budding, immunostained tumor specimens were initially scanned with a ×4 objective lens (and ×10 ocular one) to select the areas with the highest density of budding. Tumor budding in the selected areas was then counted using a ×20 objective lens and the highest count per slide was used as the number of buds. The intensity of tumor budding was arbitrarily categorized into low intensity (<5 buds/field), intermediate intensity (5–10 buds/field) and high intensity (≥10 buds/field), according to the International Tumor Budding Consensus Conference (ITBCC) 2016 recommendations [39].

CD8^+^ T-cell density was quantitatively assessed. After the CD8^+^ T-cells were identified in the tumor stroma at the periphery of the tumor at a low magnification, they were counted manually in the areas of highest CD8^+^ intensity under 400× magnification and the cell counts were averaged as previously described [40]. The histopathological evaluation of the mode of invasion at the invasive front of the tumor was conducted according to Yamamoto et al. [23]. WPOI was assessed according to previous descriptions [41].

### 4.3. Statistical Analysis

To assess the association of pathologic lymph node status and clinicopathologic variables, the chi-square test and Fisher’s exact test were performed. We used the Kaplan–Meier method to estimate OS and DSS, with the univariate difference in the survival rates being assessed by the log-rank test.

Two-tailed *p* values of <0.05 were considered to indicate statistical significance. The variables that had prognostic potential as suggested by univariate analysis were subjected to multivariate analysis with a Cox proportional hazards regression model. The model was simplified using a stepwise selection method by removing variables that were negatively associated with survival or had *p* value of greater than 0.05. Statistical analyses were performed using SPSS version 23.0 for Windows (IBM Inc., Armonk, NY, USA).

## 5. Conclusions

The results of this retrospective study show that SND (I–III) limited to the field of levels I and II for OSCC is an appropriate management technique for up to 2 pathologically positive neck lymph nodes that are no more than 3 cm in their longest diameter. Therefore, it appears unnecessary to routinely perform an MRND/RND for cN1 OSCC.

## Figures and Tables

**Figure 1 cancers-11-00269-f001:**
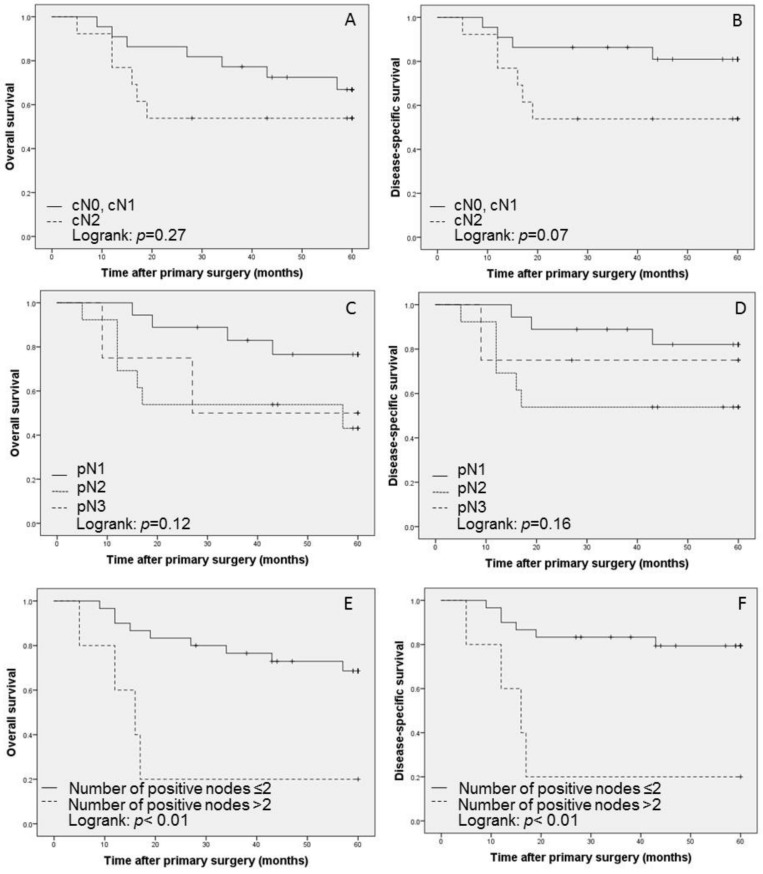
The five-year Kaplan–Meier survival estimates by extent of neck dissection for overall survival (**A**, **C** and **E**) and disease-specific survival (**B**, **D** and **F**); **A** and **B** show the survival curves of cN0, cN1 versus cN2. **C** and **D** show the survival curves of pN1 versus pN2 versus pN3. According to the number of positive lymph nodes (divided into ≤2: **E**, **F** and >2: **E**, **F**).

**Table 1 cancers-11-00269-t001:** Ipsilateral and bilateral or contralateral neck dissection according to type of neck dissection in 68 patients with pathologically node-positive oral squamous cell carcinoma.

Ipsilateral and Bilateral or Contralateral Neck Dissection		Type of Neck Dissection		
		Selective Neck Dissection	Modified Radical Neck Dissection	Radical Neck Dissection
Ipsilateral neck dissection	(*n* = 54)	31	19	4
Bi-or contralateral neck dissection	(*n* = 14)	4	9	1

**Table 2 cancers-11-00269-t002:** Level of lymph node involvement in 68 patients with pN+ and 35 pN+ who underwent SND.

Neck Side	Lymph Node Level	Total Number of Positive Necks (81 Necks ^a^)	SND (39 Necks ^a^)
Ipsilateral	I	28	23
	I + II	6	3
	I + II + III	3	0
	I + II + III + IV	1	0
	I + II + III + V	1	0
	I + II + III + IV + V	1	0
	I + III	4	0
	I + III + IV	1	0
	II	12	6
	II + III	3	0
	III	7	4
Contralateral	I	9	2
	I + II	1	0
	I + II + III + IV	2	0
	II + III	1	0
	III	1	1

pN+, pathologically node-positive; SND, selective neck dissection; ^a^, number of positive necks.

**Table 3 cancers-11-00269-t003:** Clinical variables in patients with node-positive oral squamous cell carcinoma.

Variable		Total Number of Patients with pN+ (%)	SND
No. of patients		68	35
Sex	Male	40 (59)	24
	Female	28 (19)	11
Age (years)	≤69	36 (53)	22
	>69	32 (47)	13
Site	Oral tongue	28 (41)	8
	Floor of mouth	11 (16)	8
	Upper gum	4 (6)	2
	Lower gum	15 (22)	10
	Buccal mucosa	10 (15)	7
Treatment	Surgery	38 (56)	18
	Neoadjuvant chemotherapy and/or radiotherapy plus surgery	30 (44)	17
cT classification	cT1	12 (18)	7
	cT2	32 (47)	18
	cT3	6 (9)	2
	cT4	18 (26)	8
cN classification	cN0	27 (40)	15
	cN1	10 (15)	7
	cN2	31 (45)	13
cTNM stage	I	9 (13)	5
	II	16 (24)	9
	III	7 (10)	4
	IV	36 (53)	17
Type of neck dissection	Elective	6 (9)	5
	Therapeutic	62 (91)	30
Adjuvant radiotherapy	No	56 (82)	31
	Yes	12 (18)	4
Local recurrence	No	48 (71)	24
	Yes	20 (29)	11
Regional recurrence	No	58 (85)	31
	Yes	10 (15)	4
Regional control	No	8 (12)	2
	Yes	60 (88)	33

pN+, pathologically node-positive; SND, selective neck dissection.

**Table 4 cancers-11-00269-t004:** Treatment, pathologic variables and five-year survival in 35 pN+ OSCC patients who underwent SND (I–III).

Variable		No. of Patients	Five-Year Survival			
			OS (%)	*p*-Value ^a^	DSS (%)	*p*-Value ^a^
Total		35	62	-	71	-
Treatment	Surgery	18	78	0.08	83	0.13
	Neoadjuvant chemotherapy and/or radiotherapy plus surgery	17	46		58	
Type of neck dissection	Elective	5	40	0.35	53	0.63
	Therapeutic	30	66		73	
cN classification	cN0	15	73	0.41	86	0.15
	cN1	7	54		71	
	cN2	13	54		54	
Tumor grade	Grade 1	17	76	0.19	88	0.03
	Grade 2	16	50		50	
	Grade 3	2	50		100	
pN classification	pN1	18	77	0.12	82	0.16
	pN2	13	43		54	
	pN3	4	50		75	
Highest positive anatomic level	I	20	75	0.16	85	0.11
	II	10	40		50	
	III	5	60		60	
Number of positive nodes	≤2	30	69	<0.01	79	<0.01
	>2	5	20		20	
Microscopic ENE	Absent	31	64	0.52	71	0.98
	Present	4	50		75	
Mean number of dissected lymph nodes	≤18	22	58	0.73	68	0.74
	>18	13	69		77	
LND	≤0.11	27	70	0.06	77	0.10
	>0.11	8	33		50	
Lymphovascular emboli	No	30	62	0.86	73	0.58
	Yes	5	60		60	
Perineural invasion	No	32	62	0.93	71	0.88
	Yes	3	67		67	
Surgical margin status	Negative	33	66	<0.01	75	<0.01
	Positive	2	0		0	
Tumor budding	<5	14	79	0.68	79	0.25
	5–10	16	69		56	
	≥10	5	60		40	
CD8^+^ T-cell density	≥64 cells	6	50	0.63	83	0.54
	<64 cells	27	66		70	
Mode of invasion	Grade 1 + 2 + 3	25	66	0.40	79	0.12
	Grade 4C	6	50		50	
	Grade 4D	4	50		50	
Worst pattern of invasion (WPOI)	WPOI-1 + 2 + 3 + 4	33	62	0.61	72	0.44
	WPOI-5	2	50		50	
Adjuvant radiotherapy	No	31	60	0.55	67	0.21
	Yes	4	75		100	

SND, selective neck dissection; OS, overall survival; DSS, disease-specific survival; ENE, extranodal extension; LND, lymph node density; pN+, pathologically node-positive. ^a^, Log-rank test.

**Table 5 cancers-11-00269-t005:** Multivariate Cox hazards regression model for the factors influencing overall survival and disease-specific survival in patients with node-positive squamous cell carcinoma who underwent selective neck dissection.

Variable	Factors	Overall Survival		
		HR	95% CI	*p*-Value
Before stepwise selection				
Surgical margin status	Positive	1.98	0.26–14.92	0.50
Number of positive nodes	>2	3.82	0.81–17.86	0.08
After stepwise selection				
Number of positive nodes	>2	4.98	1.48–16.72	<0.01
**Variable**	**Factors**	**Disease-Specific Survival**		
		**HR**	**95% CI**	***p*-Value**
Before stepwise selection				
Tumor grade	Grade 2 + 3	1.98	0.69–5.72	0.20
Surgical margin status	Positive	1.08	0.14–8.30	0.93
Number of positive nodes	>2	6.17	1.17–32.38	0.03
After stepwise selection				
Number of positive nodes	>2	6.44	1.76–23.50	<0.01

CI: confidence interval; HR: hazard ratio.

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
