# Peer review of "Selective Neck Dissection and Survival in Pathologically Node-Positive Oral Squamous Cell Carcinoma"

_cancers, 2019, doi:10.3390/cancers11020269_

Reviewer 1 Report

The manuscript can now be published

Author Response

Comments and Suggestions for Authors The manuscript can now be published

Response 1: We thank the reviewer for your positive comment.

Reviewer 2 Report

In this study, authors used 7th TNM classification. Therefore, the four cases with microscopic extracapsular spread (ECS) exists in pN1-2. However, as you know, in 8th TNM classification, ECS exists in pN3b. The 8th TNM classification was established in 2017. Now is already 2019. Therefore, authors should use 8th TNM classification.

Author Response

Comments and Suggestions for Authors

In this study, authors used 7th TNM classification. Therefore, the four cases with microscopic extracapsular spread (ECS) exists in pN1-2. However, as you know, in 8th TNM classification, ECS exists in pN3b. The 8th TNM classification was established in 2017. Now is already 2019. Therefore, authors should use 8th TNM classification.

Response 2: We thank the reviewer for this suggestion. According to your recommendation that we use the 8th TMN classification instead of the 7th, we changed our criteria according to Tumors of Head and Neck in the 8th version of the AJCC Cancer Staging Manual. As you pointed out, we confirmed that there were four cases with microscopic ECS in pN3b, not pN1-2; we corrected this. Additionally, we used the term of extranodal extension (ENE), which is the preferred wording (although terms such as microscopic extracapsular spread (ECS), extracapsular extension (ECE), or extranodal involvement (ENI) have also been used to denote tumor extension outside the capsule of a metastatic node, as in reference 37). On the other hand, the 8th version of the manual did not enable us to re-estimate cT classification, and thus we could not change this category. If you have further concerns, please let me know.

Reviewer 3 Report

This manuscript describes the relationship between selective neck dissection and survival based on the pathological analysis of lymph-node in oral squamous cell carcinoma. In the previous submission, it described only from a clinical viewpoint, however, the revised version has additional analysis of CD8 expression and patients' survival. Even it is not a positive relationship between the two, however, the information itself would be very valuable for the understanding of the details of patients' survival. 
I agree that this revised version would be suitable for publication from Cancers, which cause big interest, especially for young oral surgeons.

Author Response

Comments and Suggestions for Authors

This manuscript describes the relationship between selective neck dissection and survival based on the pathological analysis of lymph-node in oral squamous cell carcinoma. In the previous submission, it described only from a clinical viewpoint, however, the revised version has additional analysis of CD8 expression and patients' survival. Even it is not a positive relationship between the two, however, the information itself would be very valuable for the understanding of the details of patients' survival. 
I agree that this revised version would be suitable for publication from Cancers, which cause big interest, especially for young oral surgeons. 

Response 3: We thank the reviewer for your positive comments.

This manuscript is a resubmission of an earlier submission. The following is a list of the peer review reports and author responses from that submission.

Round  1

Reviewer 1 Report

This is a retrospective review on a small series of patients

It presents many interesting outcomes on number and levels of metastases, but unfortunately is not very clear.

There is no statistically significant difference between the MRND and SND ropes as the numbers are so small. It seems the MRND is doing worse, which reflects the fact that patients with more advanced disease were selected for that treatment. So comparison should not be done.

The only thing you could conclude is that after SND (in 35 cases), the neck control rate was 94% and becomes lower when more than 2 nodes are involved.

The results section is very unclear. 

The patient characteristics should be in a materials and methods section, and should be clarified. 

Now elective neck dissections and therapeutic are all mixed up. I would advise to exclude N3 necks (probably the RND group)

Line 77: one of 6 patients with pN+: what 6 patients ???

I count 46 SND that were pN+. Why do you just select 35 ???

I count 35 MRND/RND: why do you select 28 ??

Many different patient numbers are mentioned: 221, 131. Unclear where the selections took place to finally come to 63 ?? or 68 ?? You exclude patients with level 4-5 metastases: that is not allowed !!.

I do not see anything on postoperative RT ??? That might explain the rather high local recurrence rates. Only table 5 shows PORT, but indications are missing.

So my advise would be to write the results more clearly and not perform strange selections (excluding level 4-5 mets). Maybe best to restrict to the 46 patients that had a SND ??

Author Response

 Point 1: This is a retrospective review on a small series of patients. It presents many interesting outcomes on number and levels of metastases, but unfortunately is not very clear.

Response 1: Based on your comment, we have rewritten this text in the revised manuscript to clarify our results.

Point 2: There is no statistically significant difference between the MRND and SND ropes as the numbers are so small. It seems the MRND is doing worse, which reflects the fact that patients with more advanced disease were selected for that treatment. So comparison should not be done.

Response 2: Thank you for this pertinent suggestion. Accordingly, we have compared the clinical data of SND, excluding MRND.

Point 3: The only thing you could conclude is that after SND (in 35 cases), the neck control rate was 94% and becomes lower when more than 2 nodes are involved.

Response 3: We agree with your comment regarding our manuscript.

Point 4: The results section is very unclear. 

Response 4: Based on your comment, we have revised the complete results section to better elucidate our results.

Point 5: The patient characteristics should be in a materials and methods section, and should be clarified. 

Response 5: According to your suggestion, we moved the details regarding the patients’ characteristic to the material and methods section.

Point 6: Now elective neck dissections and therapeutic are all mixed up. I would advise to exclude N3 necks (probably the RND group)

Response 6: Thank you for your suggestion. We have excluded N3 neck.

Point 7: Line 77: one of 6 patients with pN+: what 6 patients ???

Response 7: Based on your comment, we have rewritten this text in the revised manuscript.

Point 8: I count 46 SND that were pN+. Why do you just select 35 ???

Response 8: Thank you for this comment. We have accordingly revised this.

Point 9: I count 35 MRND/RND: why do you select 28 ??

Response 9: Thank you for this comment. We have revised this in the manuscript.

Point 10: Many different patient numbers are mentioned: 221, 131. Unclear where the selections took place to finally come to 63 ?? or 68 ?? You exclude patients with level 4-5 metastases: that is not allowed !!.

Response 10: Thank you for your comment. We rewrote this in revised manuscript.

Point 11: I do not see anything on postoperative RT ??? That might explain the rather high local recurrence rates. Only table 5 shows PORT, but indications are missing.

Response 11: Thank you for this comment. We rewrote this in revised version.

Point 12: So my advise would be to write the results more clearly and not perform strange selections (excluding level 4-5 mets). Maybe best to restrict to the 46 patients that had a SND ??

Response 12: Thank you for your helpful suggestion. We focused on patients with SND and revised the text accordingly.

Reviewer 2 Report

General comments

The authors showed that SND for neck node-positive OSCC is appropriate for up to 2 lymph nodes with a maximum diameter of 3 cm or less with or without adjuvant therapy. I think that these findings are presented in detail in statistical terms and can be evaluated. However, the data of this study is not inconsistent with the data so far, but it is poor in novelty. 

Specific comments

1.    TNM classification is done in 2011 TNM Classification of Malignant Tumors (7th Edition), but should be done in the 8the Edition of 2017.

2.    Most of the data presented this time are clinical statistical data, but should also show at least the basic data that forms are basis of it.

3.    When classified into various stages, the number of N is small, and the reliability of data is lacking.

Author Response

General comments

Point 1. The authors showed that SND for neck node-positive OSCC is appropriate for up to 2 lymph nodes with a maximum diameter of 3 cm or less with or without adjuvant therapy. I think that these findings are presented in detail in statistical terms and can be evaluated. However, the data of this study is not inconsistent with the data so far, but it is poor in novelty.  

Response 1. Based on your comment, we have revised the manuscript based on their valuable feedback. Below are our point-by-point responses.

Specific comments

Point 2. TNM classification is done in 2011 TNM Classification of Malignant Tumors (7th Edition), but should be done in the 8the Edition of 2017.

Response 2. We appreciate and agree with the reviewer’s comment. At the time of diagnosis, staging of all patients was based on the 7th edition of the TNM classification. Before the study, we compared the differences in staging between the 7th and the 8th editions of the TNM classification. The main difference between the two editions was related to the classification of T2 and T3 maxillary gingival cancers, whereas the staging of T4 tumors was indifferent. In addition, obvious clinical extranodal infiltration could not be confirmed in the neck. Therefore, adoption of the 7th edition for staging should not affect the staging of the patients included in the current study.

Point 3. Most of the data presented this time are clinical statistical data, but should also show at least the basic data that forms are basis of it.

Response 3. We thank the reviewers for their review of our manuscript, insightful comments, and helpful suggestions. We have revised the manuscript based on basic data extracted from our whole database.

Point 4. When classified into various stages, the number of N is small, and the reliability of data is lacking.

Response 4. Thank you for your helpful suggestion. The number of SND is small, however, we have elucidated that SND for neck node-positive OSCC is appropriate for up to 2 lymph nodes with a maximum diameter of 3 cm or less with or without adjuvant therapy excluding MRND.

Reviewer 3 Report

This is a manuscript describing the usefulness of selective neck dissection for the treatment of head and neck cancer. They showed relatively fair results based on their treatment strategy.

The aim of the manuscript itself is not bad, however, it is not suitable for the publication from "Cancers" because results which related to basic research are missing. 

Lots of manuscripts have reported the relationship between the expression level of some genes related to oncology and clinical results even from journals which ranking is lower than "Cancers" (IF=5.326).

If authors want us to consider a possible publication from this journal, more results which show gene involvement in their clinical results, or I myself recommend them to submit other journals which focus on only clinical issues. 

Author Response

Point 1: This is a manuscript describing the usefulness of selective neck dissection for the treatment of head and neck cancer. They showed relatively fair results based on their treatment strategy. The aim of the manuscript itself is not bad, however, it is not suitable for the publication from "Cancers" because results which related to basic research are missing. Lots of manuscripts have reported the relationship between the expression level of some genes related to oncology and clinical results even from journals which ranking is lower than "Cancers" (IF=5.326).

Response 1: We thank you for this helpful comment. We have revised the manuscript for the reason the therapetic SND(I-III) is apporopriate for cN positive disease in the neck remains unclear because object measures for patient selection have not been clearly defined.

Point 2: If authors want us to consider a possible publication from this journal, more results which show gene involvement in their clinical results, or I myself recommend them to submit other journals which focus on only clinical issues. 

Response2: We thank you for this helpful comment. The standard treatment for lymph node metastasis in the neck is neck dissection. In our study, 68/131 patients (52%) had pathological node-positive metastases and 35/68 patients (51%) had selective cervical dissection. The status of cervical lymph node metastasis was well controlled in the present study, which was an important advantage of our analysis.

As you mentioned, in our study, we did not provide any prognostic factor based on basic research to predict neck metastasis. However, in our research group study, 18 OSCC cell lines and surgically resected OSCC tissues from 80 Japanese patients were analyzed for mutations in mutational hotspot regions of 50 cancer-related genes using a semiconductor-based Ion Torrent sequencer. We are planning to elucidate the correlation between tumor suppressor genes related to oncology and clinical results in basic research for pathological node-positive metastases.

Reviewer 4 Report

 In this study, authors used 7th TNM classification. Therefore, the case with microscopic ECS exists in pN1-2. However, as you know, in 8th TNM classification, ECS exists in pN3b. It is uncertain whether SND for ECS is appropriate for up to 2 lymph nodes of<3 cm in the largest diameter of the neck. Please clarify this point.

 In line 129 and 130, ECS had a significant effect on OS ans DSS(P=0.03 and 0.02), while in Table 5 I see (P=0.38 and<0.01). This is discrepancy.

Author Response

Point 1: In this study, authors used 7th TNM classification. Therefore, the case with microscopic ECS exists in pN1-2. However, as you know, in 8th TNM classification, ECS exists in pN3b. It is uncertain whether SND for ECS is appropriate for up to 2 lymph nodes of<3 cm in the largest diameter of the neck. Please clarify this point.

Response 1: Thank you for this comment. We clarified this discrepancy in our discussion section (page5, line 99).

Point 2: In line 129 and 130, ECS had a significant effect on OS ans DSS (P=0.03 and 0.02), while in Table 5 I see (P=0.38 and<0.01). This is discrepancy.

Response 2: Thank you for your comment. We rewrote this text in the revise manuscript (page5, line 103).

Round  2

Reviewer 1 Report

The manuscript has been rewritten extensively according to my suggestions and really has improved.

The stepwise selection in the tables on multivariate analysis should be explained, as it is unclear. Probably the main the cutoff of 1-2 metastases versus > 2 .

Reviewer 2 Report

It is recognized that this new version has been improved sufficiently.

Reviewer 3 Report

I am quite interested in the research which your institute is currently going on. However, your manuscript is only focused on clinical things. It is regrettable, however, it may not suitable for this journal publication. 

Reviewer 4 Report

In page5, line99, authors stated that ECS did not have a significant effect on OS and DSS in the univariate analysis. Furthermore,  in page7, line 147, authors stated that ECS does not serve as an independent prognostic factor. However, I think that the consequence is owin to postoperative irradiation, as shown in page8, line206. Therefore, I think that ECS should be contraindication for only SND. Author should state this in abstract.

For example, only SND is appropriate for up to 2 lymph nodes of<3cm without ECS, while both SND and adjuvant therapy (postoperative irradiation) is appropriate for up to 2 lymph nodes of <3cm with (micro) ECS. But, this conclusion needs the evidence. Please confuse " with or without adjuvant therapy."